

# Dynamics of hydrological model parameters: mechanisms, problems, and solution

Tian Lan[1], Kairong Lin[1,2,3], Chong−Yu Xu[4], Xuezhi Tan[1,2,3], Xiaohong Chen[1,2,3]

[1]Center for Water Resources and Environment, Sun Yat-sen University, Guangzhou, 510275, China.
[2]Guangdong Engineering Technology Research Center of Water Security Regulation and Control for Southern China, Guangzhou 510275, China.
[3]School of Civil Engineering, Sun Yat-sen University, Guangzhou, 510275, China.
[4]Department of Geosciences, University of Oslo, P.O. Box 1047, Blindern, 0316 Oslo, Norway

*Correspondence to*: Kairong Lin (linkr@mail.sysu.edu.cn)

**Abstract.** It has been demonstrated that the application of time-varying hydrological model parameters based on dynamic catchment behaviour significantly improves the accuracy and robustness of conventional models. However, the fundamental problems for calibrating dynamic parameters still need to be addressed. In this study, five calibration schemes for dynamic parameters in hydrological models were designed to investigate the underlying causes of poor model performance. The five schemes were assessed with respect to the model performance in different flow phases, the transferability of the dynamic
parameters to different time periods, the state variables and fluxes time series, and the response of the dynamic parameter set to the dynamic catchment characteristics. Furthermore, the potential reasons for the poor response of the dynamic parameter set to the catchment dynamics were investigated. The results showed that the underlying causes of poor model performance included time-invariant parameters, "compensation" among parameters, high dimensionality, and abrupt shifts in the parameters. The recommended calibration scheme exhibited good performance and overcame these problems by characterizing
the dynamic behaviour of the catchments. The main reason for the poor response of the dynamic parameter set to the catchment dynamics may be the poor convergence performance of the parameters. In addition, the assessment results of the state variables and fluxes, and the convergence performance of the parameters provided robust indications of the dominant response modes of the hydrological models in different sub-periods or catchments with distinguishing catchment characteristics.

## 1 Introduction

Hydrological modelling is an essential tool for understanding the hydrological processes of a catchment and forecasting streamflow (Liu et al., 2018; Turner et al., 2017; Delorit et al., 2017; Fenicia et al., 2014; Fenicia et al., 2018; Hublart et al., 2016; Liu et al., 2015; Höge et al., 2018; Sarrazin et al., 2016; Wi et al., 2015; Herman et al., 2013; Wagener et al., 2003; Wagener et al., 2001; Madsen, 2000). Unfortunately, the paucity of progress in model development is partly due to structural inadequacy. For example, dynamic components in hydrological models are oversimplified due to a poor understanding of their
physical mechanisms (Xiong et al., 2019; Deng et al., 2018; Dakhlaoui et al., 2017; Sarhadi et al., 2016; Pathiraja et al., 2016; Ouyang et al., 2016; Deng et al., 2016). Previous studies have demonstrated that the assumption of time-invariant parameters





is usually inappropriate. The reasons are that a unique parameter set optimized by hydrological models only represents the average hydrological processes, which do not accurately represent the dynamic response modes of the catchments processes (Pathiraja et al., 2018; Fowler et al., 2018; Zhao et al., 2017; Kim and Han, 2017; Golmohammadi et al., 2017; Delorit et al., 2017; Chen et al., 2017). To investigate the problems caused by time-invariant parameters, a control scheme, i.e., ***Scheme 1*** is

designed and assessed in this study. In this regard, the dynamics of the hydrological model parameters may be a type of compensation for models that are missing key processes such as climate- and land surface-related changes (Xiong et al., 2019; Deng et al., 2018; Wang et al., 2017b; Dakhlaoui et al., 2017; Sarhadi et al., 2016; Pathiraja et al., 2016; Ouyang et al., 2016; Deng et al., 2016; Todorovic and Plavsic, 2015).

However, a critical but often overlooked issue related to dynamic parameters is that there are linear or nonlinear correlations

among hydrological model parameters (Wagener and Kollat, 2007). For this reason, it has been conclusively demonstrated that the optimal parameters in hydrological models should not be considered as individual parameters but instead as parameter vector "teams" (Wagener and Kollat, 2007). As a result, the dynamics of the individual parameters may not represent the time-varying properties of river catchments due to the compensation between parameters (Höge et al., 2018; R. et al., 2010; Bárdossy and Singh, 2008; Bárdossy, 2007; Huang, 2005; Wagener and Kollat, 2007). Therefore, the effects of the "compensation"

between the parameters on the dynamics of hydrological model parameters are investigated using a control scheme i.e., ***Scheme 2***. The most common approach for assessing the dynamics of the hydrological model parameters is that the calibration period is partitioned into different sub-periods based on the temporal dynamic catchment characteristics (Sarhadi et al., 2016; Merz et al., 2011; Lan et al., 2018; Xiong et al., 2019; Motavita et al., 2019; Deng et al., 2018; Dakhlaoui et al., 2017; Choi and Beven, 2007; Brigode et al., 2013; Kim et al., 2015; Kim and Han, 2017; Zhao et al., 2017; Pfannerstill et al., 2015; Me et al.,

2015; Deng et al., 2016; Coron et al., 2014; Vormoor et al., 2018; Luo et al., 2012; Guse et al., 2016; Zhang et al., 2015; Ouyang et al., 2016; Zhang et al., 2011). The parameter set in each sub-period is optimized to obtain the dynamic parameter set.

Previous studies have demonstrated that sub-period calibration based on the dynamic catchment behaviour accurately captures the temporal variations of the catchment characteristics, thereby compensating for structural inadequacy (Lan et al., 2018;

Zhao et al., 2017; Kim and Han, 2017; Zhang et al., 2011; De Vos et al., 2010; Gupta et al., 2009; Choi and Beven, 2007; van Griensven et al., 2006; Freer et al., 2003). In the study of Choi and Beven (2007), the sub-periods were identified based on different hydrological characteristics using a clustering technique. The results showed that the model that considered the dynamic catchment characteristics exhibited good performance at the global level. Merz et al. (2011) demonstrated that the parameters of the catchment model related to snow and soil moisture showed clear time trends for the climate indicators. Zhang

et al. (2011) proposed a general multi-period calibration approach for improving the performance of hydrological models based on the fuzzy c-means clustering technique under time-varying climatic conditions. The results indicated that model simulations using parameters obtained from the multi-period calibration approach exhibited considerable improvements over those from the conventional single-period model. Brigode et al. (2013) demonstrated the dependence of the optimal parameter set on the





climate characteristics of the calibration period. Lan et al. (2018) applied a clustering pre-processing (CPP) framework to capture the climate-land surface variations. The results showed that the sub-annual calibration with the CPP framework exhibited significant improvements in overall performance. Even though the sub-period calibration performed well for describing the dynamics of the hydrological model parameters, some fundamental problems still need to be addressed, because

the analysis involves the hydrological model structure, global optimization, physical mechanisms of dynamic catchment characteristics, as well as complex relationships between the parameters, state variables, and fluxes. For example, multiple parameter sets are optimized simultaneously in different sub-periods. What possible disaster would be brought by parameter optimization in a high-dimensional parameter space? Therefore, in this study, a control scheme, i.e., **Scheme 3** is designed and assessed to investigate the problem of high dimensionality. Also, abrupt changes in the parameter set between two sub-periods

may result in anomalous or incorrect values in the fluxes and state variables of the time series. Hence, the control **Scheme 4** is designed to investigate potential problems caused by abrupt changes in the parameters.

These control schemes are assessed as follows: (1) the model performance is assessed at very low, low, medium, high, and very high phases of flow and the transferability of the dynamic parameter set to different time periods is determined; (2) the state variables and fluxes time series, and their changes between two consecutive sub-periods are evaluated; (3) the response

of the dynamic parameter set to the dynamic catchment characteristics is evaluated. The underlying causes for poor model performance when sub-period calibration is used are investigated and an effective calibration scheme for dynamic hydrological model parameters, **Scheme 5**, is recommended as a solution. Furthermore, the underlying mechanism of the lack of a response of the dynamic parameter set to the dynamic catchment characteristics is investigated.

The paper is structured as follows. Section 2 presents the study cases and data, the partition methods and the results of the sub-

periods based on the dynamic catchment characteristics. Section 3 elaborates on the five calibration schemes for the dynamic parameters of the hydrological model and the assessment approaches. Section 4 presents the assessment results of the different schemes, the potential problems, and the recommendation of an effective calibration scheme. Section 5 summarizes the underlying causes of poor model performance, followed by a discussion of the poor response of the dynamic parameters to the catchment dynamics. Section 6 summarizes the key conclusions of the study and outlines directions for future research.

**2 Study cases and data**

In this study, three sub-basins with different spatial scales in the Hanjiang Basin, i.e., Hanzhong basin, Mumahe basin, and Xunhe basin, were selected to demonstrate the proposed approach (Figure 1a). Climatically, the Hanjiang basin is located in the monsoon region of the East Asia subtropical zone. The area is cold and dry in winter and warm and humid in summer (Lin et al., 2010) and there are seasonal changes in vegetation density and types (Fang et al., 2002). Subtropical vegetation affects

temporal moisture conditions. Significant intra-annual changes in the climate and land-surface conditions allow for exploring the seasonal dynamics of the hydrological processes. Therefore, the three basins are ideal locations for investigating the





dynamics of hydrological model parameters. Daily streamflow and climatic data from 1980 to 1990 were used. Nearly 73% of the data samples (1980–1987) were used for calibration and the remainder (1988-1990) was utilized to verify the model.

Our previous research (Lan et al., 2018) focused on the reasonable sub-period partition based on the dynamic catchment characteristics. The study integrated data mining techniques to develop a CPP framework for sub-period partition to simulate dynamic behaviour. The hydrological model was calibrated in each sub-period to achieve the dynamics of the parameter set, as illustrated in Figure 1b. In the CPP, a set of climatic-land surface indices was provided and preprocessed using the maximal information coefficient (MIC) and principal components analysis (PCA). The climatic indices included total precipitation, maximum 1-day precipitation, maximum five-day precipitation, moderate precipitation days, heavy precipitation days, total pan evaporation, maximum 1-day pan evaporation, and minimum 1-day pan evaporation. The land-surface indices included antecedent streamflow and runoff coefficient. Two clustering operations were performed based on the preprocessed climatic index system and land-surface index system, respectively. The clustering results are shown in Figure 1c. The results showed that the performance of the model with a CPP framework was significantly improved at high, middle and low streamflow. The transferability of the dynamic parameter set from the calibration to the validation period was also greatly improved.

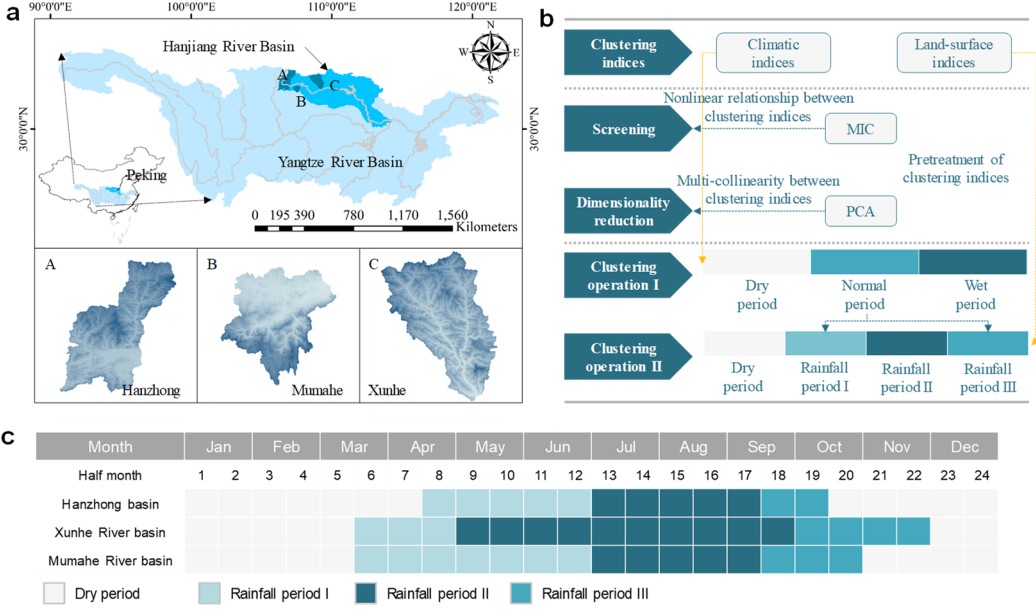

**Figure 1: Study cases and data. a,** Locations of the study region, the Hanjiang River and its three major tributaries considered in this study, i.e., the Hanzhong, Mumahe and Xunhe Rivers. **b,** Flowchart of the sub-period partition using the CPP framework. **c,** Heat map of sub-period partition.

*Note.* The sub-periods include the dry period, rainfall period I, rainfall period II, and rainfall period III. In the dry period, both the total amount and the variance values of all the precipitation series reach the minimum. In contrast, the total amount and the variance values of the precipitation series in the rainfall II (wettest period) reach the maximum and the frequency of heavy rain is highest. In the two normal sub-annual periods (rainfall period I and rainfall period III), the climatic patterns are similar but the streamflow volume is higher in rainfall period III than in rainfall period I. The reason is that higher antecedent soil moisture content contributed to the higher runoff in rainfall period III than in rainfall period I. More detailed descriptions of the clustering are described in Lan et al. (2018).



## 3 Methodology

### 3.1 Calibration schemes

The five calibration schemes are designed and compared, as follows (see Figure 2). The potential problems when dynamic of hydrological model parameters are used include time-invariant parameters, "compensation" among parameters, the high

dimensionality of the parameters, and abrupt changes in the parameters; these factors are investigated and a solution is recommended. For illustration purposes, the HYMOD model (Moore, 1985; Wagener et al., 2001; Vrugt et al., 2002; Yadav et al., 2007; De Vos et al., 2010; Pathiraja et al., 2018), which is a commonly lumped rainfall-runoff model with five parameters, is utilized. The definition of the model parameters, state variables, and fluxes are presented in Table 1. Additional information on the HYMOD model is presented in section 1 of the supporting information. All schemes with the same set of shuffled

complex evolution method were developed at the University of Arizona (SCE-UA). The SCE-UA algorithm is a subset of a global evolution algorithm (Duan et al., 1993; Hanne, 2000; Michalewicz and Schoenauer, 1996; Omran and Mahdavi, 2008; Storn and Price, 1997; Yiu-Wing and Yuping, 2001), which were used as an example of a global optimization algorithm in this study. More information is presented in section 2 of the supporting information. The simulations have a warm period of one year in the calibration period and of three months in the validation period. The objective function is defined as the

combination of the Nash-Sutcliffe efficiency index (NSE) and the logarithmic transformation (LNSE) (Nash and Sutcliffe, 1970), The NSE is sensitive to the discharge dynamics and the LNSE emphasizes the low flows because the log of the discharge is used (Nash and Sutcliffe, 1970; Guntner et al., 1999; Kiptala et al., 2014; Nijzink et al., 2016). It is expressed as

$$OF = 1 - 0.5 \cdot (\text{NSE} + \text{LNSE}) , \tag{1}$$

where $OF$ $(0, \infty)$ is the objective function value. The closer the value of $OF$ is to zero, the better the model performance.

**Scheme 1.** This scheme investigates the problem of time-invariant parameters. The parameters do not change during the entire calibration and validation periods.

**Scheme 2.** This scheme investigates the "compensation" among the parameters. In the calibration period, a specific dynamic parameter and the other fixed parameters in different sub-periods are optimized simultaneously. For example, 8 parameters, namely one specific parameter in the 4 sub-periods and another 4 fixed parameters are optimized simultaneously during one

run in HYMOD. The transition of the state variables and fluxes between two consecutive sub-periods is achieved by considering the last values of the former period as the initial values of the next period. In the validation period, the model is run using the inputs with the specific dynamic parameter and other fixed parameters. The transitions of parameters, state variables, and fluxes between two consecutive sub-periods are handled the same as in the calibration period.

The specific dynamic parameter is usually identified by whether it responds to the dynamic catchment characteristics. However,

due to the complex correlations among the parameters, the individual parameters may not represent their defined physical





characteristics. Hence, the parameter with the highest sensitivity was chosen for dynamics (Merz et al., 2011; Pfannerstill et al., 2014; Zhang et al., 2015; Deng et al., 2016; Guse et al., 2016; Ouyang et al., 2016; Deng et al., 2018; Xiong et al., 2019).

*Scheme 3.* This scheme investigates the high dimensionality of the parameters. In the calibration period, the parameter sets in different sub-periods are optimized simultaneously. For example, 20 parameters, namely 5 parameters of a hydrological model

in 4 sub-periods, are optimized simultaneously in one run. The transition of the state variables and fluxes between two consecutive sub-periods is achieved by considering the last values of the former period as the initial values of the next period. In the validation period, the model is run using the dynamic parameter set. The transitions of the parameters, state variables, and fluxes between two consecutive sub-periods are handles the same as in the calibration period.

*Scheme 4.* This scheme investigates the abrupt changes in the parameters. In the calibration period, only the data from the

individual sub-periods are used for minimizing the objective function, while the model is run for the whole period. For example, five parameters of a hydrological model in 4 sub-periods are optimized in four runs. The calibrated flow data from each sub-period are then combined and compared with the observed flow. In the validation period, the transitions of parameters, state variables, and fluxes between two consecutive sub-periods are handles the same as in the validation period of scheme 3. In the validation period, the effects of the correlations and high dimensions of the parameters are excluded and the influences caused

by the abrupt changes in the parameters are investigated.

*Scheme 5.* A solution is recommended to overcome the above problems which are caused by the time-invariant parameters, "compensation" among parameters, high dimensionality, and abrupt shifts in the parameters. In the calibration period, the model run is the same as that of the calibration period of scheme 4. In the validation period, the simulated flow data from each sub-period are combined and compared with the observed flow.

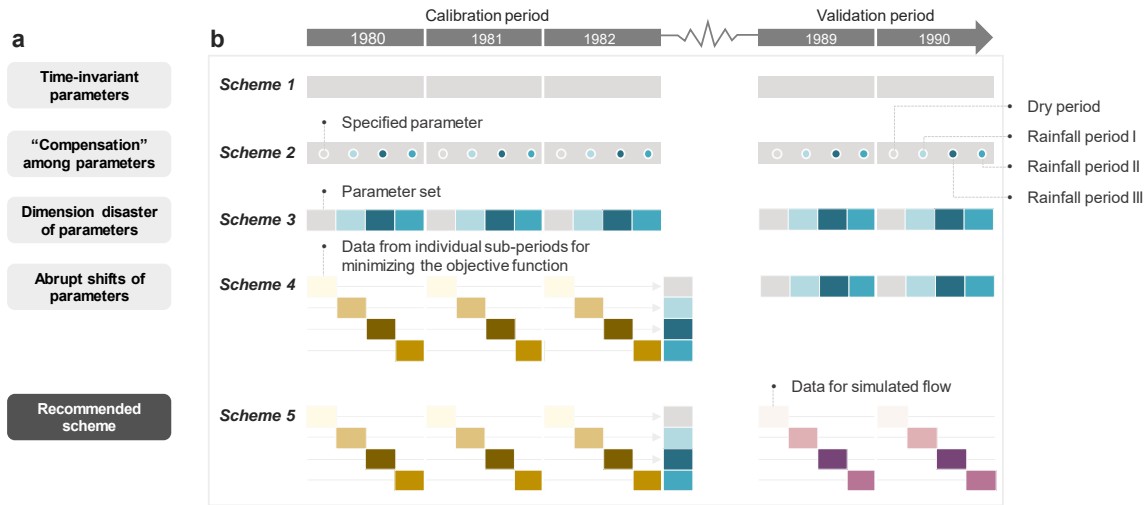

**Figure 2: Calibration schemes. a,** Objectives of the schemes. **b,** Schematic illustration of the five schemes.
*Note.* In scheme 1, the parameters are time-invariant; In scheme 2, the dynamics of only the specific parameter is operated. The specific parameters in different sub-periods and the other fixed parameters are optimized simultaneously; In scheme 3, the parameter set is dynamized.





The parameter sets in different sub-periods are optimized simultaneously; In scheme 4, only the data from the individual sub-periods are used for minimizing the objective function, while the model is run for the whole period. The parameter sets in different sub-period are optimized. In the validation period, the parameter set between two consecutive sub-periods is updated accordingly. In scheme 5, the calibration is the same as in scheme 4. In the validation period, the simulated flow data from each separate sub-period are combined and
compared with the observed flow.

**Table 1.** Definition of parameters, state variables, and fluxes used in HYMOD model (Wagener et al., 2001).

| Label | Property | Range | Description |
|---|---|---|---|
| $H_{uz}$ | Parameter | 0-1000 [mm] | Maximum height of soil moisture accounting tank |
| $B$ | Parameter | 0-1.99 | Scaled distribution function shape |
| alpha | Parameter | 0-0.99 | Quick/slow split |
| $K_q$ | Parameter | 0-0.99 | Quick-flow routing tanks' rate |
| $K_s$ | Parameter | 0-0.99 | Slow-flow routing tank's rate |
| $XH_{uz}$ | State variable | [mm] | Upper zone soil moisture tank state height |
| $XC_{uz}$ | State variable | [mm] | Upper zone soil moisture tank state contents |
| $X_q$ | State variable | [mm] | Quick-flow tank states contents |
| $X_s$ | State variable | [mm] | Slow-flow tank state contents |
| $AE$ | Flux | [mm] | Actual evapotranspiration flux |
| $OV$ | Flux | [mm] | Precipitation excess flux |
| $Q_q$ | Flux | [mm] | Quick-flow flux |
| $Q_s$ | Flux | [mm] | Slow-flow flux |
| $Q_{sim}$ | Flux | [mm] | Total streamflow flux |

### 3.2 Assessment

*Assessment of model performance.* The performance assessments of the calibration schemes include (1) an assessment of the performance in different phases of the streamflow and (2) an assessment of the transferability of the dynamic parameters to
different time. Seven performance metrics are used to assess the performance for different parts of the hydrograph in the calibration and validation periods. The metrics are listed and defined in Table 2. The differences in these metrics between the calibration period and validation period are used to assess the transferability of the optimized parameters. The transferability of the parameters to different time periods is considered a requirement for the successful validation of the model (Gharari et al., 2013; Klemeš, 1986).

**Table 2.** Definitions of the performance metrics.

| Performance metric | Description |
|---|---|
| NSE | Sensitive to peaks and discharge dynamic |
| LNSE | Emphasizing low flows with log of discharge |
| RMSE_Q5 | RMSE in FDC Q5 very low segment volume |
| RMSE_Q20 | RMSE in FDC between Q5 and Q20 low segment volume |
| RMSE_Qmid | RMSE in FDC between Q20 and Q70 mid segment volume |
| RMSE_Q70 | RMSE in FDC between Q70 and Q95 high segment volume |
| RMSE_Q95 | RMSE in FDC Q95 very high segment volume |

*Note.* The flow duration curve (FDC) is usually split into different segments to describe different flow characteristics of a catchment (Cheng et al., 2012; Coopersmith et al., 2012; Kim and Kaluarachchi, 2014; Pugliese et al., 2014; Pfannerstill et al., 2014). The RMSE with quadratic character is usually used to evaluate poor model performance due to the strong sensitivity to extreme positive and negative error values.

*Assessment of the state variables and fluxes.* The internal processes of the hydrological model run include the state variables
and fluxes time series. The abrupt changes in the parameters between two consecutive sub-periods may result in changes in



the state variables and fluxes, thereby affecting the simulation results. Hence, all the state variables and fluxes obtained from the different schemes are investigated and the underlying physical mechanisms are discussed (Kim and Han, 2017).

***Assessment of the dynamic parameter set.*** The response of the dynamic parameter sets to the dynamic catchment characteristics in all schemes is investigated for the two response modes of HYMOD, i.e., soil moisture mode and routing mode. Furthermore, the underlying physical mechanisms based on dynamic catchment characteristics are analysed.

## 4 Results

### 4.1 Model performance

The model performance of the five schemes in the Hanzhong basin is shown in Figure 3. The performance of scheme 2 is only slightly better than that of scheme 1, which indicates only a slight increase in the model performance. Scheme 3 has the worst model performance at the global level, i.e., all metrics are much higher than 1 in the calibration period and validation period. Here, the lower values of the metrics indicate better model performance. Scheme 4 has the highest overall model performance in the calibration period. For example, the NSE and LNSE are 45.3% and 13.8% respectively; these values are considerably higher than the metrics of scheme 1. The other metrics also indicate that scheme 4 performs best in all flow phases in the calibration period. However, the model performance of scheme 4 in the validation period is only slightly better than that of scheme 1. Scheme 5 has the same model performance as scheme 4 in the calibration period. Nevertheless, the overall model performance of scheme 5 is significantly higher than that of the other schemes in the validation period.

The transferability of the optimized parameters is analysed in all schemes. Scheme 5 has the smallest differences and scheme 4 has the largest differences in the metrics between the calibration period and validation period.

In summary, scheme 5 does not only have the highest overall performance under different flow conditions in the calibration period and validation period but also exhibits good transferability of the model parameters. Scheme 4 exhibits good performance in the calibration period but does not perform well in the validation period. Scheme 3 has extremely poor model performance at the global level. Scheme 2 does not have better performance than scheme 1. The evaluation results of the five schemes in the Mumahe basin and Xunhe basin are listed in section 4 of the supporting information. The results are similar to those of the Hanzhong basin and will be discussed in section 5.1.



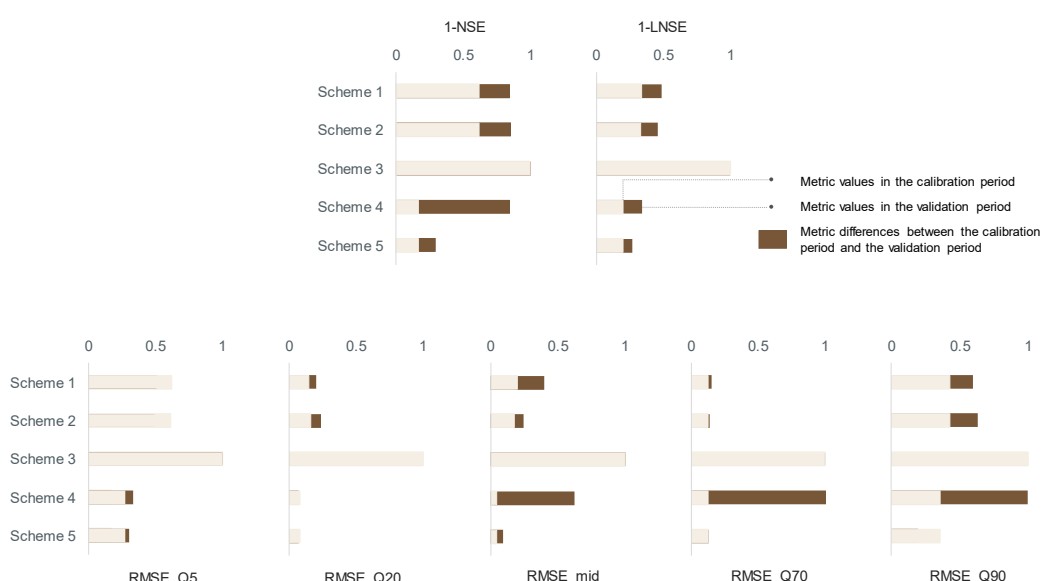

**Figure 3: Model performance.** Model performance of the five schemes in the Hanzhong basin; (1) the performance in different phases of the streamflow, and (2) the transferability of dynamic parameters to different time periods.

## 4.2 State variables and fluxes

The assessment results of the state variables and fluxes are shown in Figures 4 and 5. The variables of scheme 2 are similar to those of scheme 1. The model performance of scheme 2 is only slightly better than that of scheme 1. In scheme 3, there are some unexpected values of the state variables in the time series. In scheme 4, invalid values of the fluxes and state variables are found at the junction of sub-periods, where the parameter set exhibits abrupt changes. In scheme 5, (1) $Q_s$, $XH_{UZ}$ and $XC_{UZ}$ are lower in the dry period and higher in the rainfall period II than those in scheme 1. The results indicate that the performance

of the model run is better in the dry period and the rainfall period II because the runoff is usually overestimated in the dry period (Pool et al., 2017; Wang et al., 2017a; Tongal and Booij, 2018; Xiong et al., 2018) and underestimated in the wettest period (Guo et al., 2018; Höge et al., 2018; Pande and Moayeri, 2018; Wang et al., 2018). It is observed that the state variable $X_s$ and the flux $Q_s$ have larger effects on simulating runoff than the quick flow ($Q_q$ and $X_q$) mode in the rainfall period II. The reason is that most of the excess streamflow is diverted to the slow-flow routing, hence the fluxes and state variables present

more representative of the slow-flow tank mode. (2) A comparison of the observations and simulations of the runoff in scheme 1 and scheme 5 indicates that both peak flows in the rainfall period II are more accurately simulated by scheme 5. (3) Scheme 5 also exhibits superior performance in the two normal periods, because the state variables provide a good representation of the physical mechanism. The state variables $XH_{UZ}$ and $XC_{UZ}$ are lower in the rainfall period I and higher in the rainfall period III than in scheme 1. The reason is that the antecedent soil moisture content in the rainfall period III is higher than in the rainfall

period I (Lan et al., 2018). Consequently, the results are consistent with the results in section 4.1. The dynamic parameters in scheme 5 provided a good representation of the dynamic catchment characteristics.







**Figure 4: Result of fluxes assessment.** The fluxes (including $AE$, $OV$, $Q_q$, $Q_s$, and $Q_{sim}$) for the five schemes in the reference year in the validation period in the Hanzhong basin.

*Note.* The variables in different sub-periods are denoted by different colours (same colours as in Figure 2a. The variables of scheme 0 are denoted by the thin grey lines in each box. The observed streamflow time series data are denoted as thin red lines. All fluxes and state variables in the calibration and validation periods are presented in section 4 of the supporting information.



**Figure 5: Result of state variables assessment.** The state variables (including $XH_{UZ}$ $XC_{UZ}$ $X_{q1}$, $X_{q2}$, $X_{q3}$, and $X_s$) for the five schemes in the reference year in the validation period in the Hanzhong basin.





### 4.3 Dynamic parameter set

The dynamic parameter values optimized by the four sub-period calibration schemes in the Hanzhong basin are shown in Figure 6. In scheme 2, the dynamic parameter $K_q$ with the highest identifiability and the other fixed parameters are optimized. The result shows that the responses of the dynamic parameter $K_q$ to the dynamic catchment characteristics are not clear. In

scheme 3, the parameters $H_{UZ}$ and $B$ in the soil moisture mode of HYMOD (Moore, 1985; Vrugt et al., 2002) show no regular patterns in any of the schemes and this is similar for $\alpha$, $K_q$, and $K_s$ in the slow- and quick-flow routing mode. In short, the dynamic parameters do not show clear responses to the dynamic catchment characteristics in scheme 2 or scheme 3. In scheme 5, which is the same as scheme 4 in the calibration period, $K_s$ accurately describes the model responses in the sub-periods for the different catchment characteristics. The value of $K_s$ is lowest in the dry period and highest in the wettest period. However,

the parameter $K_q$ exhibits no significantly regular changes. The main reason is that most of the excess streamflow in the three rainfall periods is diverted to the slow-flow tank because the $\alpha$ values are close to zero. This means that the quick-flow tanks do not have an effect on the simulations. The parameter sets optimized by scheme 1 and scheme 5 in the Mumahe basin and Xunhe basin are listed in section 4 of supporting information. The results are similar to those of the Hanzhong basin.

In summary, scheme 5 performs best for identifying the dominant parameters and their responses to the dynamic catchment

characteristics. The dynamic features of the parameters also demonstrate the necessity for sub-period calibration. Furthermore, it is interesting that the state variables and fluxes describe the dynamic catchment behaviour more robustly than the dynamic parameters. In light of this, the underlying causes for the poor response of the dynamic parameter set to the catchment dynamics are investigated.

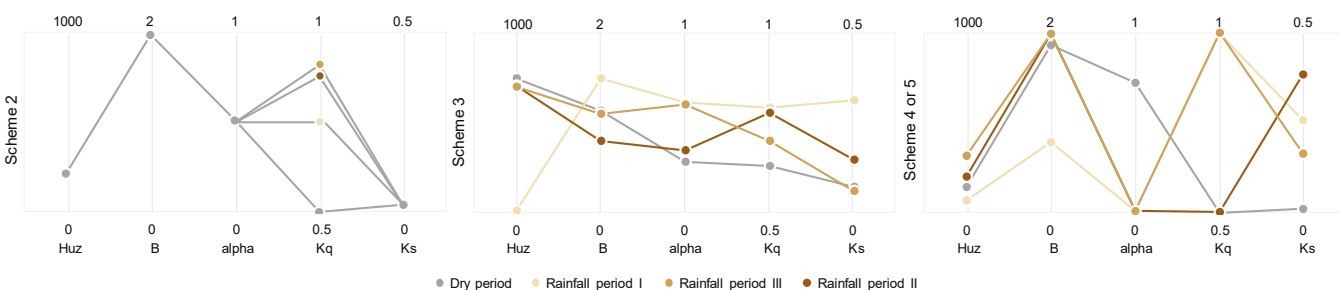

**Figure 6: Result of dynamic parameter set assessment.** The dynamic parameter sets optimized by the four sub-period calibration schemes in the Hanzhong basin.

## 5 Discussion

### 5.1 Underlying causes of poor model performance

The evaluation results of the five schemes are summarized to explore the possible reasons for poor model performance:





1.  **Time-invariant parameters:** The scheme 1 with the time-invariant parameter set averages the hydrological responses. As a result, scheme 1 resulted in a poor simulation accuracy or weak transferability of the optimized parameters in different flow conditions. The results were consistent with Delorit et al. (2017), Fowler et al. (2018) and Xiong et al. (2019).

5  2.  **"Compensation" among parameter:** In scheme 2, the individual parameters with high identifiability did show clear responses to the dynamic catchment characteristics. Bárdossy (2007) demonstrated that changes in one parameter may be compensated for by changes in other parameters due to their interdependence (Westra et al., 2014; Klotz et al., 2017; Wang et al., 2017b; Wang et al., 2018). Therefore, although a specific parameter is dynamic, the other parameters may counteract those changes, resulting in no overall change in the hydrological processes. Hence, the model performance in scheme 2 was relatively low.

3.  **High dimensionality of parameters:** In scheme 3, it has a sound logic by continuously running the model with the dynamic parameter set like the real system. However, the results showed that all fluxes and state variables in the time series were anomalous and the model exhibited extremely poor performance. It was demonstrated that parameter optimization in high-dimensional parameter space with correlated parameters resulted in the failure of the modelling run (Beven and Binley, 1992; Sivakumar, 2004; Bárdossy and Singh, 2008; Laloy and Vrugt, 2012).

4.  **Abrupt shifts in parameters:** In scheme 4, the abrupt shifts in the parameter set between the sub-periods resulted in anomalous values in the fluxes and state variables in the time series, which results in failure of the model in the validation period. Kim and Han (2017) also emphasized the negative effects of abrupt shifts in the parameter set on the model performance.

20  In summary, scheme 5 is recommended for dynamic hydrological model parameters because it can capture the temporal variations of the dynamic catchment characteristics and overcome the underlying problems responsible for poor model performance. Although scheme 5 has higher computational cost, this does not represent a large problem with current computing devices.

**5.2 Underlying causes of the poor response of the dynamic parameters to the catchment dynamics**

25  The dynamic parameter set was estimated using global optimization algorithms. However, if the convergence fails, the global optimum cannot be determined and the optimal parameter values may be anomalous. In this case, the optimal results do not represent the hydrological processes in a catchment. (Gomez, 2019; Weise, 2009). In order to investigate the underlying causes of the poor response of the dynamic parameters to the catchment dynamics, we assessed the convergence performance of the dynamic parameters and determine the ability of the parameters to respond to the catchment dynamics (Zecchin et al., 2012; 30  Zheng et al., 2017; Azad and Optimization, 2019).



### 5.2.1 A tool for the convergence evaluation of the dynamic parameters

In order to overcome the limitation of traditional tools for evaluating the convergence behaviour of global optimization algorithms for hydrological models, including the visualization of the high-dimensional parameter response surface, rough response surfaces with discontinuous derivatives, poor or inconsistent sensitivities of the response surface, non-convex mesh
surfaces and the dynamic convergence process in high-dimensional parameter spaces (Duan et al., 1992; Sorooshian et al., 1993; Duan et al., 1994; Cooper et al., 1997; Gupta et al., 1998; Vrugt et al., 2005; Weise, 2009; Zhang et al., 2009; Sun et al., 2012; Arora and Singh, 2013; Derrac et al., 2014; Piotrowski et al., 2017; Gomez, 2019), a simple and powerful approach is proposed, namely, the Evaluate the Convergence Performance using Violin Plots (ECP-VP). This tool represents the potential features of the fitness landscapes (see Figure 7) and provides a visualization of the convergence behaviour in multi-parameter
space. The strategy is as follows.

1.    The end of each evolution loop in the optimization process is regarded as a cut-off point. The parameter set with the best objective function value in each evolution loop is recorded in the "convergence process".

2.    Violin plots, which are an excellent tool to visualize the kernel density distribution of the data points (Hintze and Nelson, 1998; Piel et al., 2010), are used to configure the convergence process in the individual parameter spaces. The probability
distributions of the violin plots are used to represent the possible properties of the fitness landscapes. The anatomy of the violin plot and the associated information can be found in section 3 of the supporting information. With an adequate parameter space and sufficient density of coverage, the four types of distributions of violin plots are matched to the property' sketches of the fitness landscapes (Weinberger, 1990; Forrest, 1995; Harik et al., 1999; Gibbs et al., 2004; Arsenault et al., 2014; Maier et al., 2014).

3.    A decrease in the performance of the convergence and the candidate mechanisms are interpreted as (I) a unimodal distribution: an ideal global convergence process is used to estimate the best solution. The unimodal distribution matches two types of fitness landscape sketches including the best case and low variation. (II) Bimodal distribution: there are two main local optima and the distance to the two local convergence regions is far. It becomes more complicated for the optimization process to find the global optimum and the premature convergence to a local optimum may occur (Duan et
al., 1992; 1993; 1994; Weise, 2009; Sun et al., 2012; Derrac et al., 2014; Gomez, 2019). The bimodal distribution symbolizes the two types of fitness landscape sketches including the multimodal and deceptive types. (III) Multimodal distribution: the response surface may be multi-modal plus steep ascends and descends. This means that multiple local optima exist. With the maze of minor local optima, the calibration algorithm may fail to reach the global optimum. Because the minor optima may be found quite far from the global optimum, the search may terminate prematurely without
finding an approximate solution (Dakhlaoui et al., 2017; Duan et al., 1994; Duan et al., 1993; Duan et al., 1992). The multimodal distribution matches the three types of fitness landscape sketches including the multimodal, rugged, and deceptive types. (IV) Flat distribution: this is similar to the multimodal distribution and its surface may be noisy. The very poor sensitivity of the objective function to the parameter fluctuation causes weak convergence of the parameter




(Duan et al., 1992; Duan et al., 1993; Duan et al., 1994; Dakhlaoui et al., 2017; Rahnamay Naeini et al., 2018; Vrugt and Beven, 2018). The flat distribution matches the three types of fitness landscape sketches including the neutral, needle-in-a-haystack, and nightmare types.

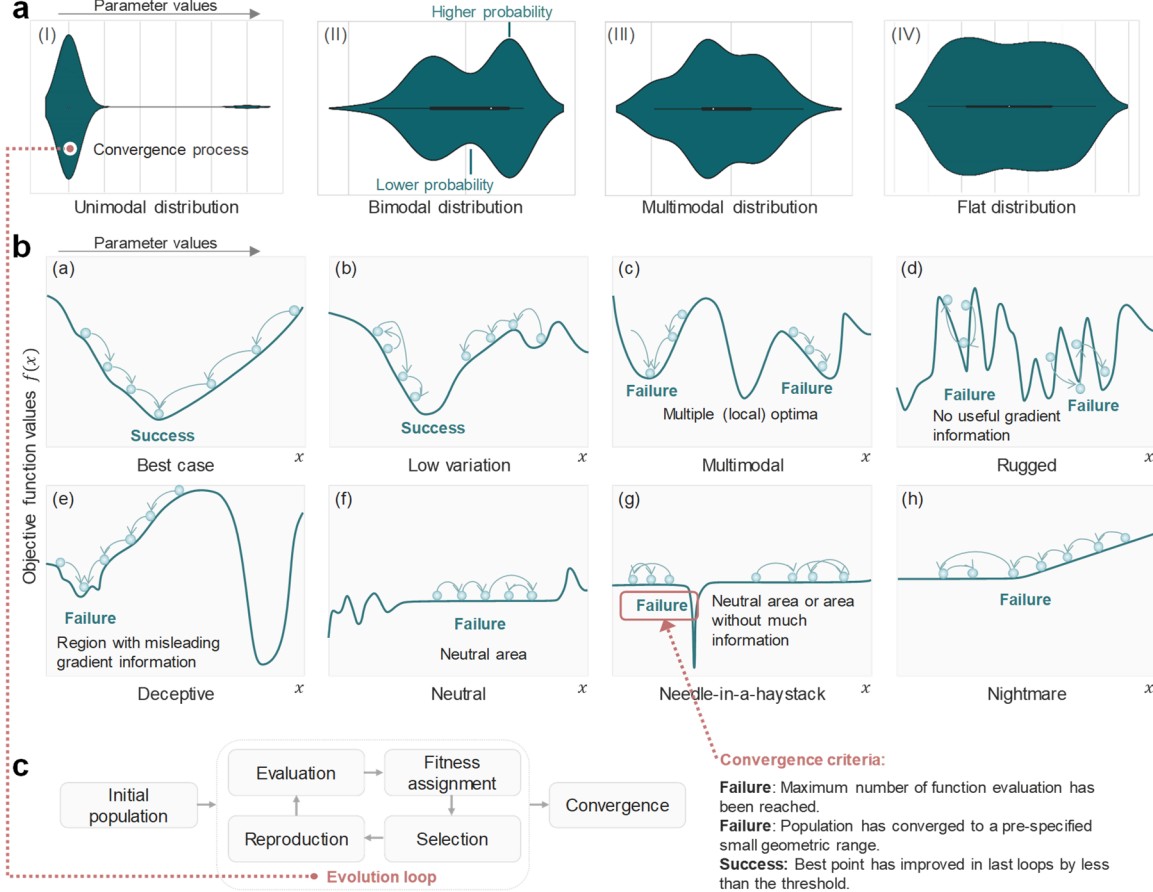

**Figure 7: Evaluation of the ability of the dynamic parameters to response to the catchment dynamics. a,** Evaluation of the convergence processes using violin plots (ECP-VP). The horizontal axis of the violin plot denotes the parameter values and the vertical axis denotes the probability values. The probability distribution of elements of the search space is represented by the violin plots. **b,** All possible properties of the fitness landscapes. **c,** The basic cycle of the global evolution algorithm. Initial population: create an initial population of random individuals; evaluation: compute the objective values of the solution candidates; fitness assignment: use the objective values to determine
the fitness values; selection: select the fittest individuals for reproduction; reproduction: create new individuals from the mating pool by crossover and mutation.
*Note.* Fitness landscapes are a very powerful metaphor for visualizing the convergence processes in global optimization. Some intuitive sketches of fitness landscapes with possible properties are as follows. The horizontal axis denotes the parameter values and the vertical axis denotes the objective function values. The direction of the arrow represents the direction of evolution. The possible properties include (a)
best case: an optimization process is ideal for estimating the globally optimal parameters. (b) Low variation: an optimization process with low variation is fair for estimating the globally optimal parameters. (c) Premature convergence: an optimization process has prematurely converged to a local optimum if it is no longer able to explore other parts of the search space than the area currently being examined and there exists another region that contains a superior solution. (d) Ruggedness: if the objective function values are fluctuating, i.e., increasing or decreasing, it is difficult to determine the correct direction for the optimization process. In short, ruggedness is multi-modality plus steep
ascends and descends in the fitness landscape. (e) Deceptiveness: the gradient of the deceptive objective function values leads the optimizer



away from the optima. (f) Neutrality: the outcome of the application of a search operation to an element of the search space is neutral if it yields no change in the objective function values. (g) Needle-In-A-Haystack: the optimum occurs as an isolated spike in a plane, representing the occurrences of extreme ruggedness combined with a general lack of information in the fitness landscape. (h) Nightmare: the optimum is difficult to achieve in an approximate plane. More details on the fitness landscapes and their properties can be found int Weise (2009).

### 5.2.2 Convergence assessment

The convergence assessment results of scheme 1 and scheme 5 in the Hanzhong basin are shown in Figure 8. In scheme 1, (1) the parameter $B$ represents the bimodal distribution in the parameter space, indicating that the fitness landscape of $B$ is unsteady or fluctuating (see Figure 8 (a)). It is inferred that the convergence processes of the parameter $B$ may be affected by a prominent local optimum. The outcomes of the search operations may be arbitrary, which leads to a divergence away from the global optima. As a result, the convergence performance of $B$ is poor. (2) Although $H_{UZ}$, $\alpha$, $K_q$, and $K_s$ rapidly converge and the range is small, the magnification (Figure 8 (b)) shows bimodal or multimodal distributions. The global optima cannot be determined in the $H_{UZ}$, $\alpha$, $K_q$, and $K_s$ parameter space. As a consequence, the convergence of the parameters in scheme 1 is poor and the response of the parameters to the catchment behaviour with a low level of confidence.

In scheme 5, the four sub-periods are evaluated separately. (1) In the dry period, except for $K_s$, the distributions of the other parameters are oscillating in the entire feasible parameter space. Indeed, the magnification of parameter $K_s$ (see Figure 8 (b)) shows a multimodal distribution. The result indicates that the convergence performance of the parameter set in the dry period is poor. Due to the weak relationship between precipitation and runoff in the dry period (Moore, 1985; Yadav et al., 2007; De Vos et al., 2010), most modules of the model in the dry period do not accurately characterize the behaviour of the catchment. (2) In the rainfall periods I, II, and III, the parameters $\alpha$ and $K_s$ with unimodal distribution have the best convergence performance. The $\alpha$ values in the three rainfall periods are close to the minimum, hence the slow-flow tank controls the cascade routing component of the model. The $\alpha$ and $K_s$ with high identicality and best convergence performance also demonstrate that the chosen model is most suitable for the streamflow simulation in the three rainfall periods. The main reason is that the HYMOD model is well suited for catchments dominated by "saturation excess overland flow" processes. Intense rainfall events contribute to saturation excess overland flow in the rainfall periods (Herman et al., 2013; Sarrazin et al., 2016; Wang et al., 2017a; Wang et al., 2018). Moreover, the results also illustrate that the optimal $\alpha$ and $K_q$ (or $K_s$) in the cascade routing component have higher reliability than the optimal $H_{UZ}$ and $B$ in the soil moisture component. In summary, the dynamic parameters $\alpha$ and $K_q$ (or $K_s$) in scheme 5 have good convergence performance and accurately describe the response to the dynamic catchment characteristics. However, the parameters $H_{UZ}$ and $B$ with poor convergence performance exhibit a poor ability to describe the response to the dynamic catchment characteristics. Interestingly, the convergence performance results of the parameters for the dominant response modes in HYMOD are consistent with the results of the performance of the state variables and fluxes and the dynamic parameter set. The evaluation results of ECP-VP for scheme 1 and scheme 5 in the Mumahe basin and Xunhe basin are shown in section 5 of the supporting information. The results are similar to those of the Hanzhong basin.



**Figure 8: Convergence performance for scheme 1 and scheme 5 in the Hanzhong basin. (a)** The convergence processes in the parameter spaces; **(b)** magnification of the convergence processes of the parameters.



The following results were observed: (1) the proposed ECP-VP tool accurately described the convergence behaviour of the models in the individual parameter spaces, demonstrating the reliability of the optimized dynamic parameter values to response to dynamic catchment characteristics. The tool can be used to determine the reason for the potentially poor convergence performance. (2) The convergence performance can be used to identify the operation modes of hydrological models and

provides valuable guidance for the improvement of hydrological models with different catchment characteristics. (3) The convergence performance of the parameters in one sub-period might be superior or inferior to those of other sub-periods. For example, the convergence performance of all parameters was worse in the dry period than in the three rainfall periods. Indeed, due to the complex correlations between the parameters in a parameter set, the convergence performance of an individual parameter may be significantly affected by the other parameters. For this reason, it is not recommended to use the convergence

performance of individual parameters but rather the convergence performance of the parameter set. However, the application of this solution requires a significant amount of experiments, validation, analysis, and discussion and these points will be investigated in future studies.

## 6 Conclusions

We designed five calibration schemes for the dynamics of hydrological model parameters to investigate the underlying causes

of poor model performance. An assessment system was proposed to determine an appropriate calibration scheme. The potential reasons for the poor response of the dynamic parameter set to the catchment dynamics were discussed. The following conclusions were drawn:

1.    The five schemes were systematically evaluated with respect to the model performance in different flow phases, the transferability of the dynamic parameters to different time periods, the state variable and flux time series, and the response

20        of the dynamic parameter set to the dynamic catchment characteristics. The possible reasons for the poor model performance included (1) time-invariant parameters, (2) "compensation" among parameters, (3) high dimensionality of the parameters, and (4) abrupt shifts of the parameters. Interestingly, the results also proved that changes in the state variables and fluxes time series provided a more robust description of the dynamic catchment characteristics than the dynamic parameters.

2.    The proposed calibration (1) compensated for the deficiencies in the model structure, (2) provided high forecast accuracy for different flow phases, (3) exhibited good transferability of the model parameters between the calibration and validation periods, (4) improved the ability to identify the dominant parameters and their responses to the catchment processes, (5) accurately characterized the dynamic behaviour of the catchments.

3.    The reasons for the poor response of the dynamic parameter to the catchment dynamics were determined by assessing

30        the convergence performance of the dynamic parameters. The results indicated that the dynamic parameters with good convergence performance accurately described the response to the dynamic catchment characteristics, whereas





Brigode, P., Oudin, L., and Perrin, C.: Hydrological model parameter instability: A source of additional uncertainty in estimating the hydrological impacts of climate change?, J Hydrol, 476, 410-425, 10.1016/j.jhydrol.2012.11.012, 2013.

Chen, Y., Chen, X. W., Xu, C. Y., Zhang, M. F., Liu, M. B., and Gao, L.: Toward improved calibration of SWAT using season-based multi-objective optimization: A case study in the Jinjiang basin in southeastern China, Water Resour Manag, 32, 1193-1207, https://doi.org/10.1007/s11269-017-1862-8, 2017.

Cheng, L., Yaeger, M., Viglione, A., Coopersmith, E., Ye, S., and Sivapalan, M.: Exploring the physical controls of regional patterns of flow duration curves – Part 1: Insights from statistical analyses, Hydrol Earth Syst Sc, 16, 4435-4446, 10.5194/hess-16-4435-2012, 2012.

Choi, H. T., and Beven, K.: Multi-period and multi-criteria model conditioning to reduce prediction uncertainty in an application of TOPMODEL within the GLUE framework, J Hydrol, 332, 316-336, 10.1016/j.jhydrol.2006.07.012, 2007.

Cooper, V. A., Nguyen, V. T. V., and Nicell, J. A.: Evaluation of global optimization methods for conceptual rainfall-runoff model calibration, Water Science and Technology, 36, 53-60, https://doi.org/10.1016/S0273-1223(97)00461-7, 1997.

Coopersmith, E., Yaeger, M. A., Ye, S., Cheng, L., and Sivapalan, M.: Exploring the physical controls of regional patterns of flow duration curves – Part 3: A catchment classification system based on regime curve indicators, Hydrol Earth Syst Sc, 16, 4467-4482, 10.5194/hess-16-4467-2012, 2012.

Coron, L., Andreassian, V., Perrin, C., Bourqui, M., and Hendrickx, F.: On the lack of robustness of hydrologic models regarding water balance simulation: a diagnostic approach applied to three models of increasing complexity on 20 mountainous catchments, Hydrol Earth Syst Sc, 18, 727-746, 10.5194/hess-18-727-2014, 2014.

Dakhlaoui, H., Ruelland, D., Tramblay, Y., and Bargaoui, Z.: Evaluating the robustness of conceptual rainfall-runoff models under climate variability in northern Tunisia, J Hydrol, 550, 201-217, 10.1016/j.jhydrol.2017.04.032, 2017.

Dawkins, R.: Climbing mount improbable, WW Norton & Company, 1997.

de Vos, N. J., Rientjes, T. H. M., and Gupta, H. V.: Diagnostic evaluation of conceptual rainfall-runoff models using temporal clustering, Hydrol Process, 24, 2840-2850, 10.1002/hyp.7698, 2010.

Delorit, J., Ortuya, E. C. G., and Block, P.: Evaluation of model-based seasonal streamflow and water allocation forecasts for the Elqui Valley, Chile, Hydrol Earth Syst Sc, 21, 4711-4725, https://doi.org/10.5194/hess-21-4711-2017, 2017.

Deng, C., Liu, P., Guo, S. L., Li, Z. J., and Wang, D. B.: Identification of hydrological model parameter variation using ensemble Kalman filter, Hydrol Earth Syst Sc, 20, 4949-4961, https://doi.org/10.5194/hess-20-4949-2016, 2016.

Deng, C., Liu, P., Wang, D. B., and Wang, W. G.: Temporal variation and scaling of parameters for a monthly hydrologic model, Journal of Hydrology, 558, 290-300, https://doi.org/10.1016/j.jhydrol.2018.01.049, 2018.

Derrac, J., García, S., Hui, S., Suganthan, P. N., and Herrera, F.: Analyzing convergence performance of evolutionary algorithms: A statistical approach, Information Sciences, 289, 41-58, https://doi.org/10.1016/j.ins.2014.06.009, 2014.

Duan, Q. Y., Gupta, V. K., and Sorooshian, S.: Shuffled Complex Evolution Approach for Effective and Efficient Global Minimization, Journal of Optimization Theory and Applications, 76, 501-521, Doi 10.1007/Bf00939380, 1993.



Duan, Q., Sorooshian, S., and Gupta, V. K.: Optimal use of the SCE-UA global optimization method for calibrating watershed models, Journal of hydrology, 158, 265-284, 1994.

Duan, Q., Sorooshian, S., and Gupta, V.: Effective and efficient global optimization for conceptual rainfall-runoff models, Water Resources Research, 28, 1015-1031, 10.1029/91WR02985, 1992.

Eckhardt, K., and Arnold, J. G.: Automatic calibration of a distributed catchment model, Journal of Hydrology, 251, 103-109, https://doi.org/10.1016/s0022-1694(01)00429-2, 2001.

Fenicia, F., Kavetski, D., Reichert, P., and Albert, C.: Signature-Domain Calibration of Hydrological Models Using Approximate Bayesian Computation: Empirical Analysis of Fundamental Properties, Water Resources Research, 54, 3958-3987, 10.1002/2017wr021616, 2018.

Forrest, T. J. a. S.: Fitness Distance Correlation as a Measure of Problem Difficulty for Genetic Algorithms, Proceedings of the Sixth International Conference on Genetic Algorithms, 184--192, 1995.

Fowler, K., Coxon, G., Freer, J., Peel, M., Wagener, T., Western, A., Woods, R., and Zhang, L.: Simulating Runoff Under Changing Climatic Conditions: A Framework for Model Improvement, Water Resources Research, 0, doi:10.1029/2018WR023989, 2018.

Freer, J., Beven, K., and Peters, N.: Multivariate seasonal period model rejection within the generalised likelihood uncertainty estimation procedure, Calibration of watershed models, 69-87, http://refhub.elsevier.com/S1364-8152(15)00173-5/sref28, 2003.

Gavrilets, S.: Fitness landscapes and the origin of species (MPB-41), Princeton University Press, 2004.

Gharari, S., Hrachowitz, M., Fenicia, F., and Savenije, H. H. G.: An approach to identify time consistent model parameters:
20       sub-period calibration, Hydrol Earth Syst Sc, 17, 149-161, 10.5194/hess-17-149-2013, 2013.

Gibbs, M. S., Maier, H. R., and Dandy, G. C.: Applying fitness landscape measures to water distribution optimization problems, in: Hydroinformatics, World Scientific Publishing Company, 795-802, 2004.

Golmohammadi, G., Rudra, R., Dickinson, T., Goel, P., and Veliz, M.: Predicting the temporal variation of flow contributing areas using SWAT, Journal of Hydrology, 547, 375-386, https://doi.org/10.1016/j.jhydrol.2017.02.008, 2017.

Gomez, J.: Stochastic global optimization algorithms: A systematic formal approach, Information Sciences, 472, 53-76, https://doi.org/10.1016/j.ins.2018.09.021, 2019.

Guntner, A., Uhlenbrook, S., Seibert, J., and Leibundgut, C.: Multi-criterial validation of TOPMODEL in a mountainous catchment, Hydrological Processes, 13, 1603-1620, https://doi.org/10.1002/(sici)1099-1085(19990815)13:11<1603::aid-hyp830>3.3.co;2-b, 1999.

Guo, D., Johnson, F., and Marshall, L.: Assessing the Potential Robustness of Conceptual Rainfall-Runoff Models Under a Changing Climate, Water Resources Research, 54, 5030-5049, doi:10.1029/2018WR022636, 2018.



Gupta, H. V., Kling, H., Yilmaz, K. K., and Martinez, G. F.: Decomposition of the mean squared error and NSE performance criteria: Implications for improving hydrological modelling, Journal of Hydrology, 377, 80-91, https://doi.org/10.1016/j.jhydrol.2009.08.003, 2009.

Gupta, H. V., Sorooshian, S., and Yapo, P. O.: Toward improved calibration of hydrologic models: Multiple and noncommensurable measures of information, Water Resources Research, 34, 751-763, doi:10.1029/97WR03495, 1998.

Guse, B., Pfannerstill, M., Strauch, M., Reusser, D. E., Ludtke, S., Volk, M., Gupta, H., and Fohrer, N.: On characterizing the temporal dominance patterns of model parameters and processes, Hydrol Process, 30, 2255-2270, 10.1002/hyp.10764, 2016.

Hanne, T. J. J. o. H.: Global Multiobjective Optimization Using Evolutionary Algorithms, 6, 347-360, 10.1023/a:1009630531634, 2000.

Harik, G., Cantú-Paz, E., Goldberg, D. E., and Miller, B. L.: The Gambler's Ruin Problem, Genetic Algorithms, and the Sizing of Populations, 7, 231-253, 10.1162/evco.1999.7.3.231, 1999.

Herman, J. D., Reed, P. M., and Wagener, T.: Time-varying sensitivity analysis clarifies the effects of watershed model formulation on model behavior, Water Resources Research, 49, 1400-1414, 10.1002/wrcr.20124, 2013.

Hintze, J. L., and Nelson, R. D.: Violin plots: a box plot-density trace synergism, The American Statistician, 52, 181-184, https://doi.org/10.2307/2685478, 1998.

Höge, M., Wöhling, T., and Nowak, W.: A primer for model selection: The decisive role of model complexity, Water Resources Research, 54, 1688-1715, 2018.

Huang, G. H.: Model identifiability, Wiley StatsRef: Statistics Reference Online, 2005.

Hublart, P., Ruelland, D., De Cortázar-Atauri, L. G., Gascoin, S., Lhermitte, S., and Ibacache, A.: Reliability of lumped hydrological modeling in a semi-arid mountainous catchment facing water-use changes, Hydrol. Earth Syst. Sci., 20, 3691-3717, https://doi.org/10.5194/hess-20-3691-2016, 2016.

Kauffman, S. A.: The origins of order: Self-organization and selection in evolution, OUP USA, 1993.

Khakbaz, F., and Kazeminezhad, M.: Work hardening and mechanical properties of severely deformed AA3003 by constrained groove pressing, Journal of Manufacturing Processes, 14, 20-25, https://doi.org/10.1016/j.jmapro.2011.07.001, 2012.

Kim, D., and Kaluarachchi, J.: Predicting streamflows in snowmelt-driven watersheds using the flow duration curve method, Hydrol Earth Syst Sc, 18, 1679-1693, 10.5194/hess-18-1679-2014, 2014.

Kim, K. B., and Han, D.: Exploration of sub-annual calibration schemes of hydrological models, Hydrology Research, 48, 1014-1031, 10.2166/nh.2016.296, 2017.

Kim, K. B., Kwon, H.-H., and Han, D.: Hydrological modelling under climate change considering nonstationarity and seasonal effects, Hydrology Research, nh2015103, 10.2166/nh.2015.103, 2015.



Kiptala, J. K., Mul, M. L., Mohamed, Y. A., and van der Zaag, P.: Modelling stream flow and quantifying blue water using a modified STREAM model for a heterogeneous, highly utilized and data-scarce river basin in Africa, Hydrol Earth Syst Sc, 18, 2287-2303, https://doi.org/10.5194/hess-18-2287-2014, 2014.

Klemeš, V.: Operational testing of hydrological simulation models, Hydrological Sciences Journal, 31, 13-24, 1986.

Klotz, D., Herrnegger, M., and Schulz, K.: Symbolic Regression for the Estimation of Transfer Functions of Hydrological Models, Water Resources Research, 53, 9402-9423, 10.1002/2017wr021253, 2017.

Laloy, E., and Vrugt, J. A.: High-dimensional posterior exploration of hydrologic models using multiple-try DREAM(ZS) and high-performance computing, Water Resources Research, 48, 10.1029/2011wr010608, 2012.

Lan, T., Lin, K. R., Liu, Z. Y., He, Y. H., Xu, C. Y., Zhang, H. B., and Chen, X. H.: A Clustering Preprocessing Framework for the Subannual Calibration of a Hydrological Model Considering Climate-Land Surface Variations, Water Resources Research, 54, 10,034-010,052, 10.1029/2018wr023160, 2018.

Liu, Z. Y., Cheng, L. Y., Hao, Z. C., Li, J. J., Thorstensen, A., and Gao, H. K.: A framework for exploring joint effects of conditional factors on compound floods, Water Resour. Res., https://doi.org/10.1002/2017WR021662, 2018.

Liu, Z. Y., Zhou, P., Chen, X. Z., and Guan, Y. H.: A multivariate conditional model for streamflow prediction and spatial precipitation refinement, J. Geophys. Res. Atmos., 120, https://doi.org/0.1002/2015JD02378, 2015.

Luo, J. M., Wang, E. L., Shen, S. H., Zheng, H. X., and Zhang, Y. Q.: Effects of conditional parameterization on performance of rainfall-runoff model regarding hydrologic non-stationarity, Hydrol Process, 26, 3953-3961, 10.1002/hyp.8420, 2012.

Madsen, H.: Automatic calibration of a conceptual rainfall–runoff model using multiple objectives, Journal of Hydrology, 235, 276-288, https://doi.org/10.1016/S0022-1694(00)00279-1, 2000.

Maier, H. R., Kapelan, Z., Kasprzyk, J., Kollat, J., Matott, L. S., Cunha, M. C., Dandy, G. C., Gibbs, M. S., Keedwell, E., Marchi, A. J. E. M., and Software: Evolutionary algorithms and other metaheuristics in water resources: Current status, research challenges and future directions, 62, 271-299, 2014.

Me, W., Abell, J. M., and Hamilton, D. P.: Effects of hydrologic conditions on SWAT model performance and parameter sensitivity for a small, mixed land use catchment in New Zealand, Hydrol Earth Syst Sc, 19, 4127-4147, 10.5194/hess-19-4127-2015, 2015.

Merz, R., Parajka, J., and Bloschl, G.: Time stability of catchment model parameters: Implications for climate impact analyses, Water Resources Research, 47, 10.1029/2010wr009505, 2011.

Michalewicz, Z., and Schoenauer, M.: Evolutionary Algorithms for Constrained Parameter Optimization Problems, Evolutionary Computation, 4, 1-32, 10.1162/evco.1996.4.1.1, 1996.

Moore, R. J.: The probability-distributed principle and runoff production at point and basin scales, Hydrological Sciences Journal, 30, 273-297, 10.1080/02626668509490989, 1985.

Motavita, D. F., Chow, R., Guthke, A., and Nowak, W.: The comprehensive differential split-sample test: A stress-test for hydrological model robustness under climate variability, J Hydrol, 573, 501-515, 10.1016/j.jhydrol.2019.03.054, 2019.





Nash, J. E., and Sutcliffe, J. V.: River flow forecasting through conceptual models part I - A discussion of principles, Journal of Hydrology, 10, 282-290, https://doi.org/10.1016/0022-1694(70)90255-6, 1970.

Nijzink, R. C., Samaniego, L., Mai, J., Kumar, R., Thober, S., Zink, M., Schäfer, D., Savenije, H. H. G., and Hrachowitz, M.: The importance of topography-controlled sub-grid process heterogeneity and semi-quantitative prior constraints in
distributed hydrological models, Hydrol Earth Syst Sc, 20, 1151-1176, https://doi.org/10.5194/hess-20-1151-2016, 2016.

Omran, M. G. H., and Mahdavi, M.: Global-best harmony search, Applied Mathematics and Computation, 198, 643-656, 10.1016/j.amc.2007.09.004, 2008.

Ouyang, Y., Xu, D., Leininger, T. D., and Zhang, N.: A system dynamic model to estimate hydrological processes and water use in a eucalypt plantation, Ecological Engineering, 86, 290-299, 10.1016/j.ecoleng.2015.11.008, 2016.

Pande, S., and Moayeri, M.: Hydrological Interpretation of a Statistical Measure of Basin Complexity, Water Resources Research, 54, 7403-7416, doi:10.1029/2018WR022675, 2018.

Pathiraja, S., Anghileri, D., Burlando, P., Sharma, A., Marshall, L., and Moradkhani, H.: Time-varying parameter models for catchments with land use change: the importance of model structure, Hydrol Earth Syst Sc, 22, 2903-2919, 10.5194/hess-22-2903-2018, 2018.

Pathiraja, S., Marshall, L., Sharma, A., and Moradkhani, H.: Hydrologic modeling in dynamic catchments: A data assimilation approach, Water Resources Research, 52, 3350-3372, https://doi.org/10.1002/2015wr017192, 2016.

Pfannerstill, M., Guse, B., and Fohrer, N.: Smart low flow signature metrics for an improved overall performance evaluation of hydrological models, Journal of Hydrology, 510, 447-458, https://doi.org/10.1016/j.jhydrol.2013.12.044, 2014.

Pfannerstill, M., Guse, B., Reusser, D., and Fohrer, N.: Process verification of a hydrological model using a temporal parameter
sensitivity analysis, Hydrol Earth Syst Sc, 19, 4365-4376, 10.5194/hess-19-4365-2015, 2015.

Piel, F. B., Patil, A. P., Howes, R. E., Nyangiri, O. A., Gething, P. W., Williams, T. N., Weatherall, D. J., and Hay, S. I.: Global distribution of the sickle cell gene and geographical confirmation of the malaria hypothesis, Nat Commun, 1, 104, https://doi.org/10.1038/ncomms1104, 2010.

Piotrowski, A. P., Napiorkowski, M. J., Napiorkowski, J. J., and Rowinski, P. M.: Swarm Intelligence and Evolutionary
Algorithms: Performance versus speed, Information Sciences, 384, 34-85, https://doi.org/10.1016/j.ins.2016.12.028, 2017.

Pool, S., Viviroli, D., and Seibert, J.: Prediction of hydrographs and flow-duration curves in almost ungauged catchments: Which runoff measurements are most informative for model calibration?, Journal of Hydrology, 554, 613-622, 10.1016/j.jhydrol.2017.09.037, 2017.

Pugliese, A., Castellarin, A., and Brath, A.: Geostatistical prediction of flow–duration curves in an index-flow framework, Hydrol Earth Syst Sc, 18, 3801-3816, 10.5194/hess-18-3801-2014, 2014.

R., C., P., S. K., and I., C.: Sensitivity and identifiability of stream flow generation parameters of the SWAT model, Hydrological Processes, 24, 1133-1148, doi:10.1002/hyp.7568, 2010.



Rahnamay Naeini, M., Yang, T., Sadegh, M., AghaKouchak, A., Hsu, K.-l., Sorooshian, S., Duan, Q., and Lei, X.: Shuffled Complex-Self Adaptive Hybrid EvoLution (SC-SAHEL) optimization framework, Environ Modell Softw, 104, 215-235, 10.1016/j.envsoft.2018.03.019, 2018.

Sarhadi, A., Burn, D. H., Concepción Ausín, M., and Wiper, M. P.: Time-varying nonstationary multivariate risk analysis using a dynamic Bayesian copula, Water Resources Research, 52, 2327-2349, 10.1002/2015wr018525, 2016.

Sarrazin, F., Pianosi, F., and Wagener, T.: Global Sensitivity Analysis of environmental models: Convergence and validation, Environ Modell Softw, 79, 135-152, https://doi.org/10.1016/j.envsoft.2016.02.005, 2016.

Sivakumar, B.: Dominant processes concept in hydrology: moving forward, Hydrological Processes, 18, 2349-2353, 10.1002/hyp.5606, 2004.

Sorooshian, S., Duan, Q., and Gupta, V. K.: Calibration of rainfall-runoff models: Application of global optimization to the Sacramento Soil Moisture Accounting Model, Water Resources Research, 29, 1185-1194, 10.1029/92wr02617, 1993.

Storn, R., and Price, K.: Differential Evolution – A Simple and Efficient Heuristic for global Optimization over Continuous Spaces, Journal of Global Optimization, 11, 341-359, 10.1023/a:1008202821328, 1997.

Sun, J., Wu, X., Palade, V., Fang, W., Lai, C.-H., and Xu, W.: Convergence analysis and improvements of quantum-behaved particle swarm optimization, Information Sciences, 193, 81-103, https://doi.org/10.1016/j.ins.2012.01.005, 2012.

Todorovic, A., and Plavsic, J.: The role of conceptual hydrologic model calibration in climate change impact on water resources assessment, Journal of Water and Climate Change, 7, jwc2015086, 10.2166/wcc.2015.086, 2015.

Tongal, H., and Booij, M. J.: Simulation and forecasting of streamflows using machine learning models coupled with base flow separation, Journal of Hydrology, 564, 266-282, https://doi.org/10.1016/j.jhydrol.2018.07.004, 2018.

Turner, S. W. D., Bennett, J. C., Robertson, D. E., and Galelli, S.: Complex relationship between seasonal streamflow forecast skill and value in reservoir operations, Hydrol. Earth Syst. Sci., 21, 4841-4859, https://doi.org/10.5194/hess-21-4841-2017, 2017.

van Griensven, A., Meixner, T., Grunwald, S., Bishop, T., Diluzio, M., and Srinivasan, R.: A global sensitivity analysis tool for the parameters of multi-variable catchment models, J Hydrol, 324, 10-23, 10.1016/j.jhydrol.2005.09.008, 2006.

Vormoor, K., Heistermann, M., Bronstert, A., and Lawrence, D.: Hydrological model parameter (in)stability – "crash testing" the HBV model under contrasting flood seasonality conditions, Hydrological Sciences Journal, 63, 991-1007, 10.1080/02626667.2018.1466056, 2018.

Vrugt, J. A., and Beven, K. J.: Embracing equifinality with efficiency: Limits of Acceptability sampling using the DREAM (LOA) algorithm, Journal of Hydrology, 559, 954-971, 10.1016/j.jhydrol.2018.02.026, 2018.

Vrugt, J. A., Bouten, W., Gupta, H. V., and Sorooshian, S.: Toward improved identifiability of hydrologic model parameters: The information content of experimental data, Water Resources Research, 38, 48-41-48-13, doi:10.1029/2001WR001118, 2002.



Vrugt, J. A., Diks, C. G. H., Gupta, H. V., Bouten, W., and Verstraten, J. M.: Improved treatment of uncertainty in hydrologic modeling: Combining the strengths of global optimization and data assimilation, Water Resources Research, 41, 10.1029/2004wr003059, 2005.

Wagener, T., and Kollat, J.: Numerical and visual evaluation of hydrological and environmental models using the Monte Carlo analysis toolbox, Environ Modell Softw, 22, 1021-1033, https://doi.org/10.1016/j.envsoft.2006.06.017, 2007.

Wagener, T., Boyle, D. P., Lees, M. J., Wheater, H. S., Gupta, H. V., and Sorooshian, S.: A framework for development and application of hydrological models, Hydrol. Earth Syst. Sci., 5, 13-26, 10.5194/hess-5-13-2001, 2001.

Wagener, T., McIntyre, N., Lees, M. J., Wheater, H. S., and Gupta, H. V.: Towards reduced uncertainty in conceptual rainfall-runoff modelling: Dynamic identifiability analysis, Hydrol Process, 17, 455-476, 10.1002/hyp.1135, 2003.

Wang, S., Ancell, B., Huang, G., and Baetz, B.: Improving Robustness of Hydrologic Ensemble Predictions Through Probabilistic Pre-and Post-Processing in Sequential Data Assimilation, Water Resources Research, 54, 2129-2151, 2018.

Wang, S., Huang, G. H., Baetz, B. W., and Ancell, B. C.: Towards robust quantification and reduction of uncertainty in hydrologic predictions: Integration of particle Markov chain Monte Carlo and factorial polynomial chaos expansion, Journal of Hydrology, 548, 484-497, 10.1016/j.jhydrol.2017.03.027, 2017a.

Wang, S., Huang, G. H., Baetz, B. W., Cai, X. M., Ancell, B. C., and Fan, Y. R.: Examining dynamic interactions among experimental factors influencing hydrologic data assimilation with the ensemble Kalman filter, Journal of Hydrology, 554, 743-757, 10.1016/j.jhydrol.2017.09.052, 2017b.

Weinberger, E. J. B. C.: Correlated and uncorrelated fitness landscapes and how to tell the difference, 63, 325-336, 10.1007/bf00202749, 1990.

Weise, T.: Global optimization algorithms-theory and application, Self-published, 2, 2009.

Westra, S., Thyer, M., Leonard, M., Kavetski, D., and Lambert, M.: A strategy for diagnosing and interpreting hydrological model nonstationarity, Water Resources Research, 50, 5090-5113, 2014.

Wi, S., Yang, Y. C. E., Steinschneider, S., Khalil, A., and Brown, C. M.: Calibration approaches for distributed hydrologic models in poorly gaged basins: implication for streamflow projections under climate change, Hydrol. Earth Syst. Sci., 19, 857-876, 10.5194/hess-19-857-2015, 2015.

Wright, S.: The roles of mutation, inbreeding, crossbreeding, and selection in evolution, na, 1932.

Xiong, B., Xiong, L., Chen, J., Xu, C.-Y., and Li, L.: Multiple causes of nonstationarity in the Weihe annual low-flow series, Hydrol Earth Syst Sc, 22, 1525-1542, 10.5194/hess-22-1525-2018, 2018.

Xiong, M., Liu, P., Cheng, L., Deng, C., Gui, Z., Zhang, X., and Liu, Y.: Identifying time-varying hydrological model parameters to improve simulation efficiency by the ensemble Kalman filter: A joint assimilation of streamflow and actual evapotranspiration, Journal of Hydrology, 568, 758-768, https://doi.org/10.1016/j.jhydrol.2018.11.038, 2019.



Yadav, M., Wagener, T., and Gupta, H.: Regionalization of constraints on expected watershed response behavior for improved predictions in ungauged basins, Advances in Water Resources, 30, 1756-1774, https://doi.org/10.1016/j.advwatres.2007.01.005, 2007.

Yiu-Wing, L., and Yuping, W.: An orthogonal genetic algorithm with quantization for global numerical optimization, Ieee T Evolut Comput, 5, 41-53, 10.1109/4235.910464, 2001.

Zecchin, A. C., Simpson, A. R., Maier, H. R., Marchi, A., and Nixon, J. B.: Improved understanding of the searching behavior of ant colony optimization algorithms applied to the water distribution design problem, 48, 10.1029/2011wr011652, 2012.

Zhang, D. J., Chen, X. W., Yao, H. X., and Lin, B. Q.: Improved calibration scheme of SWAT by separating wet and dry seasons, Ecol Model, 301, 54-61, 10.1016/j.ecolmodel.2015.01.018, 2015.

Zhang, H., Huang, G. H., Wang, D. L., and Zhang, X. D.: Multi-period calibration of a semi-distributed hydrological model based on hydroclimatic clustering, Advances in Water Resources, 34, 1292-1303, 10.1016/j.advwatres.2011.06.005, 2011.

Zhang, X., Srinivasan, R., Zhao, K., and Liew, M. V.: Evaluation of global optimization algorithms for parameter calibration of a computationally intensive hydrologic model, Hydrological Processes, 23, 430-441, 10.1002/hyp.7152, 2009.

Zhao, B. R., Dai, H. C., Han, D. W., and Rong, G. W.: The sub-annual calibration of hydrological models considering climatic intra-annual variations, Hydrology and Earth System Sciences Discussions, 1-15, https://doi.org/10.5194/hess-2017-396, 2017.

Zheng, F., Zecchin, A. C., Newman, J. P., Maier, H. R., and Dandy, G. C.: An Adaptive Convergence-Trajectory Controlled Ant Colony Optimization Algorithm With Application to Water Distribution System Design Problems, Ieee T Evolut Comput, 21, 773-791, 10.1109/TEVC.2017.2682899, 2017.