# Peer review of "Dynamics of hydrological model parameters: mechanisms, problems, and solution"

_Hydrology and Earth System Sciences, 2019_

## Referee Comment (RC1) · Anonymous Referee #1 · 26 Dec 2019

**Reviews of HESS-2019-544**

**Title:** Dynamics of hydrological model parameters: calibration and reliability

As fresh read of the revised paper, I think the paper reads well for most of aspects:  paper structure, descriptions, focus, figures. Reviewing the responses, the authors addressed my comments adequately.  I appreciate the revisions made based on my comments, but I still feel that introduction could be improved (though I do see the effort to revise that section).  I think one more moderate revision would improve the paper.

**Comment on introduction**

$1^{st}$ paragraph: I think this paragraph is fine. A key message is that time-invariant parameter is not appropriate. Optimized hydrologic model (based the period covering various climate conditions) represent average hydrologic dynamics over the time used for optimization.  Root of the problem is model structure deficiency (missing processes etc.).

$2^{nd}$ paragraph:  This paragraph states two different things – 1) parameter dependencies and 2) introduction of the sub-period calibration strategy (which is raised based on the problem statement in the $1^{st}$ paragraph).  I have hard time linking 1) and 2), though there is a sentence (line 12-14) that tries to link.   I agree to the point 1) that parameters should be treated as a set of parameter not individual parameter independently, for regionalization, etc.  I feel that topic of the first half of the paragraph is abrupt.

$3^{rd}$ paragraph: This paragraph describes the previous studies on sub-period calibration studies in the first half and some problem statements (issue on high dimension calibration caused by use of multiple calibration periods).

Reading $2^{nd}$ and $3^{rd}$ paragraphs carefully, I would like to suggest combing two paragraphs (last half of $2^{nd}$ paragraph and first half of $3^{rd}$ paragraph are duplicated) and state the issues stated in $2^{nd}$ paragraph and $3^{rd}$ paragraph (which this paper is trying to investigate) together.  Also it would be nice to state what the issues in the previous sub-period calibration strategies were (currently the paper states "the previous studies using sub-period calibration improved the performance, so what is the problem/question left?). I think this is needed to introduce the schemes used in this paper.

$4^{th}$ paragraph: this paragraph states how each scheme is assessed.  I am not sure if there is need for detailed evaluation methods (L12-16), but I think I am ok with this paragraph.

**Other comments:**

Thank you for describing the method for partitioning of the simulation period into sub-periods. Here the authors provided additional information, which trigger my questions. "*The results showed that the performance of the model with a CPP framework was significantly improved at*

*high, middle and low streamflow. The transferability of the dynamic parameter set from the calibration to the validation period was also greatly improved.*".  This sentence gives nuance that sub-period calibration done in the previous study was successful. So, what really motivated this current study based on your previous study?  I would suggest stating what issue/questions was posed in the previous study that motivates this current study, and what the differences in calibration methods between current paper and the previous paper are.  This could be stated in introduction?

Also, thanks for responding to my question on selection of the dynamic parameter (that is exposed to sub-period calibration in the scheme 2)?  I would suggest stating which parameter (among 5 HYMOD parameters) is the dynamic parameter (or indicating it in table 2).   To me, the selection of the dynamic parameters should be based on temporal changes in soil (might be negligible) and land cover (may be significant), but I think I understand some of bucket and flux parameters are sensitive to climate condition due to imperfect model structures (missing processes or oversimplified parameterizations in the model).

**Very minor line-by-line comments:**

P2. L13.  Spell out R. et al., 2010.

P2. L28.  At the global level -> overall period (or similar)?

P 5. L9.  I feel the model schematic figure of the model would be worth adding in main text (in addition to Table 1). This would help reader to interpret the figure 4-5 better.  Please consider.

P 5. L13-14.  This sentence (*The simulation have a warm period…*) is not clear.

P8. L11.  Suggest move this sentence (*Here, the lower values….*) immediate after the first sentence of the paragraph.

Figure 4.  I cannot tell difference between Scheme 4 and Scheme 5. I thought this would be error. I expect noticeable differences between the two schemes. But not sure.  Please check.

---

## Referee Comment (RC2) · Anonymous Referee #2 · 2 Jan 2020

Please address the following issues:

1 The introduction should have a paragraph stating the objectives of the paper.

2 In the methodology, it is unclear what happens with the dynamic parameters during the validation period. Are they set to the same values as in the calibration period? Do they follow the exact same dynamics? Are the values dependent on the calendar day?

3 Correct the units in Table 1 (fluxes).

4 In the results, the term "model performance" is very generic. Can it be replaced with something more specific?

[Figure]

544, 2019.

---

## Author Comment (AC1) · 10 Jan 2020

Dear Anonymous Referee #1,

Re: Manuscript # HESS-2019-544 entitled "Dynamics of hydrological model parameters: mechanisms, problems, and solution".

Many thanks for your positive evaluation and encouragement for the results and scientific significance of this study. We greatly appreciate the Referee's comments, especially in the focus of the first goal and textural improvements. All suggestions are helpful to improve this manuscript.

We have carefully studied and considered all comments in making revision and a point-by-point response is as follows. For clarity, all comments are given in black

and responses are given in the blue text. The revised parts in our manuscript are highlighted in red.

Please also note the supplement to this comment:
https://www.hydrol-earth-syst-sci-discuss.net/hess-2019-544/hess-2019-544-AC1-supplement.pdf

**Supplement:**

**Replies to Referee #1**

Dear Anonymous Referee #1,

Re: Manuscript # HESS-2019-544 entitled "Dynamics of hydrological model parameters: mechanisms, problems, and solution".

Many thanks for your positive evaluation and encouragement for the results and scientific significance of this study. We greatly appreciate the Referee's comments, especially in the focus of the first goal and textural improvements. All suggestions are helpful to improve this manuscript.

We have carefully studied and considered all comments in making revision and a point-by-point response is as follows. For clarity, all comments are given in black and responses are given in the blue text. The revised parts in our manuscript are highlighted in red.

Yours sincerely,

Kairong Lin (Ph.D.)
Professor in hydrology
E-mail: linkr@mail.sysu.edu.cn

**Title: Dynamics of hydrological model parameters: mechanisms, problems, and solution**

As fresh read of the revised paper, I think the paper reads well for most of aspects: paper structure, descriptions, focus, figures. Reviewing the responses, the authors addressed my comments adequately. I appreciate the revisions made based on my comments, but I still feel that introduction could be improved (though I do see the effort to revise that section). I think one more moderate revision would improve the paper.

**Reply:** We sincerely appreciate the Referee's positive evaluation and encouragement. With your constructive suggestions, the second revised version has paid more focus to improving the introduction.

**Comment on introduction:**

1st paragraph: I think this paragraph is fine. A key message is that time-invariant parameter is not appropriate. Optimized hydrologic model (based the period covering various climate conditions) represent average hydrologic dynamics over the time used for optimization. Root of the problem is model structure deficiency (missing processes etc.).

**Reply:** Many thanks for your positive evaluation.

2nd paragraph: This paragraph states two different things – 1) parameter dependencies and 2) introduction of the sub-period calibration strategy (which is raised based on the problem statement in the 1st paragraph). I have hard time linking 1) and 2), though there is a sentence (line 12-14) that tries to link. I agree to the point 1) that parameters should be treated as a set of parameter not individual parameter independently, for regionalization, etc. I feel that topic of the first half of the paragraph is abrupt.

3rd paragraph: This paragraph describes the previous studies on sub-period calibration studies in the first half and some problem statements (issue on high dimension calibration caused by use of multiple calibration periods).

Reading 2nd and 3rd paragraphs carefully, I would like to suggest combing two paragraphs (last half of 2nd paragraph and first half of the 3rd paragraph are duplicated) and state the issues stated in 2nd paragraph and 3rd paragraph (which this paper is trying to investigate) together. Also it would be nice to state what the issues in the previous sub-period calibration strategies were (currently the paper states "the previous studies using sub-period calibration improved the performance, so what is the problem/question left?). I think this is needed to introduce the schemes used in this paper.

**Reply:** Thank you for your comment and advice, following which we have done the following: The 2nd paragraph is reorganized. Its logic and links of the text are strengthened. The second half the 2nd paragraph is moved to the 3rd paragraph. The explicit information is as follows:

"However, a critical but often overlooked issue related to dynamic parameters is that there are linear or nonlinear correlations among hydrological model parameters, also called the "compensation" between parameters (Wagener and Kollat, 2007). The compensation between parameters could even result in the dynamics of the individual parameters may not represent the time-varying properties of river catchments (Höge et al., 2018; Cibin et al., 2010; Bárdossy and Singh, 2008; Bárdossy, 2007; Huang, 2005; Wagener and Kollat, 2007). Hence, it has been conclusively demonstrated that the optimal parameters in hydrological

models should not be considered as individual parameters but instead as parameter vector "teams" (Wagener and Kollat, 2007). In this research, the effects of the "compensation" between the parameters on the dynamics of hydrological model parameters are investigated using a control scheme i.e., ***Scheme 2***."

Although the previous studies using sub-period calibration improved the performance, however, the problems left and the introduction of schemes in this study are as follows:

"Even though the sub-period calibration performed well for describing the dynamics of the hydrological model parameters, some fundamental problems still need to be addressed, because the analysis involves the hydrological model structure, global optimization, physical mechanisms of dynamic catchment characteristics, as well as complex relationships between the parameters, state variables, and fluxes. For example, multiple parameter sets are optimized simultaneously in different sub-periods. Question like 'what possible disaster would be brought by parameter optimization in a high-dimensional parameter space' remains to be answered. This study aims to investigate the underlying causes of poor model performance in hydrological models with dynamic parameters via designing five calibration schemes, and explore the potential reasons for the poor response of the dynamic parameter set to the catchment dynamics are explored. In addition to schemes 1 & 2 described above, this study designed and assessed a control scheme, i.e., ***Scheme 3*** to investigate the problem of high dimensionality. Also, abrupt changes in the parameter set between two sub-periods may result in anomalous or incorrect values in the fluxes and state variables of the time series. Hence, control ***Scheme 4*** is designed to investigate potential problems caused by abrupt changes in the parameters."

4[th] paragraph: this paragraph states how each scheme is assessed. I am not sure if there is need for detailed evaluation methods (L12-16), but I think I am ok with this paragraph.
**Reply:** We appreciate that the Referee is in favor of the content of this research.

**Other comments:**
Thank you for describing the method for partitioning of the simulation period into sub-periods. Here the authors provided additional information, which trigger my questions. "*The results showed that the performance of the model with a CPP framework was significantly improved at high, middle and low streamflow. The transferability of the dynamic parameter set from the calibration to the validation period was also greatly improved.*". This sentence gives nuance that sub-period calibration done in the previous study was successful. So, what really motivated this current study based on your previous study? I would suggest stating what issue/questions was posed in the previous study that motivates this current study, and what the differences in calibration methods between current paper and the previous paper are. This could be stated in introduction?
**Reply:** We have better explained that our previous study focused on the sub-period clustering or partition based on the climate-land surface variations and relevant studies, such as the choice and preprocess of clustering indices in the light of various catchment characteristics, the clustering operation based on different clustering index systems.

However, some fundamental problems for dynamics of hydrological model parameters still need to be addressed, because the analysis involves the hydrological model structure,

global optimization, physical mechanisms of dynamic catchment characteristics, as well as complex relationships between the parameters, state variables, and fluxes. Hence, in this study, we focus on investigating the underlying causes of poor model performance in hydrological models with dynamic parameters via designing five calibration schemes. The best scheme is recommended as a solution by assessing the model performance in multiple facets, including different phases of the hydrograph precisely, the transferability of the dynamic parameters to different time periods, the state variables and fluxes time series, and the response of the dynamic parameter set to the dynamic catchment characteristics. Furthermore, the potential reasons for the poor response of the dynamic parameter set to the catchment dynamics are explored.

The above statement is added in the *Introduction* section of the revision.

Also, thanks for responding to my question on selection of the dynamic parameter (that is exposed to sub-period calibration in the scheme 2)? I would suggest stating which parameter (among 5 HYMOD parameters) is the dynamic parameter (or indicating it in table 2). To me, the selection of the dynamic parameters should be based on temporal changes in soil (might be negligible) and land cover (may be significant), but I think I understand some of bucket and flux parameters are sensitive to climate condition due to imperfect model structures (missing processes or oversimplified parameterizations in the model).

**Reply:** Thanks for your constructive comment on this point. We agree with the Referee's view in the selection of the dynamic parameters. The specific information is supplemented in the revision and as follows:

"The specific dynamic parameter is usually identified by whether it responds to the dynamic catchment characteristics. However, due to the complex correlations among the parameters and imperfect model structures (missing processes or oversimplified parameterizations in the model), the individual parameters may not represent their defined physical characteristics, such as temporal changes in soil, land cover and climate conditions. Hence, the parameter with the highest sensitivity was chosen as dynamic parameter (Merz et al., 2011; Pfannerstill et al., 2014; Zhang et al., 2015; Deng et al., 2016; Guse et al., 2016; Ouyang et al., 2016; Deng et al., 2018; Xiong et al., 2019). In this study, the dynamic parameter $K_q$ with the highest identifiability and the other fixed parameters are optimized. The chosen parameter is marked in Table 1."

**Very minor line-by-line comments:**
P2. L13. Spell out R. et al., 2010.
**Reply:** Revised as suggested.

P2. L28. At the global level -> overall period (or similar)?
**Reply:** Thanks for your reminding. The more specific explanation is supplemented as suggested.
"The results showed that the model that considered the dynamic catchment characteristics exhibited good performance at the global level (i.e., overall calibration and validation periods)."

P 5. L9. I feel the model schematic figure of the model would be worth adding in main text (in addition to Table 1). This would help reader to interpret the figure 4-5 better. Please consider.

**Reply:** Thanks for Referee's advice. The additional information on the HYMOD model is added in the main text as suggested.

P 5. L13-14. This sentence (The simulation have a warm period…) is not clear.

**Reply:** The unclear sentence is revised and its detailed information is as follows:

"The simulations have a warm-up period of one year in the calibration period and of three months in the validation period."

P8. L11. Suggest move this sentence (Here, the lower values….) immediate after the first sentence of the paragraph.

**Reply:** Revised as suggested.

Figure 4. I cannot tell difference between Scheme 4 and Scheme 5. I thought this would be error. I expect noticeable differences between the two schemes. But not sure. Please check.

**Reply:** We have better explained that the model run in the calibration period is the same in scheme 4 and scheme 5. However, the model run in the validation period is actually different. In the validation period of scheme 4, the model runs one time using the dynamic parameter set. The parameter set between two consecutive sub-periods is switched. As a result, the transition of the state variables and fluxes between two consecutive sub-periods is abrupt and achieved by considering the last values of the former period as the initial values of the next period. In the validation period of scheme 5, the model runs $N$ times ($N$ is the number of the divided sub-periods) combining the simulated flow data in the sub-periods, respectively. The comparison between scheme 4 and scheme 5 is to investigate the effect of the abrupt shifts in the parameters on the model run with dynamic parameters. The corresponding supplements have been stated in the description of scheme 5.

**References:**

Bárdossy, A., and Singh, S.: Robust estimation of hydrological model parameters, Hydrol Earth Syst Sc, 12, 1273-1283, 2008.

Bárdossy, A.: Calibration of hydrological model parameters for ungauged catchments, Hydrol. Earth Syst. Sci., 11, 703-710, 10.5194/hess-11-703-2007, 2007.

Cibin, R., Sudheer, K. P., and Chaubey, I.: Sensitivity and identifiability of stream flow generation parameters of the SWAT model, Hydrol Process, 24, 1133-1148, 10.1002/hyp.7568, 2010.

Deng, C., Liu, P., Guo, S. L., Li, Z. J., and Wang, D. B.: Identification of hydrological model parameter variation using ensemble Kalman filter, Hydrol Earth Syst Sc, 20, 4949-4961, https://doi.org/10.5194/hess-20-4949-2016, 2016.

Deng, C., Liu, P., Wang, D. B., and Wang, W. G.: Temporal variation and scaling of parameters for a monthly hydrologic model, Journal of Hydrology, 558, 290-300, https://doi.org/10.1016/j.jhydrol.2018.01.049, 2018.

Guse, B., Pfannerstill, M., Strauch, M., Reusser, D. E., Ludtke, S., Volk, M., Gupta, H., and Fohrer, N.: On characterizing the temporal dominance patterns of model parameters and processes, Hydrol Process, 30, 2255-2270, 10.1002/hyp.10764, 2016.

Höge, M., Wöhling, T., and Nowak, W.: A primer for model selection: The decisive role of model complexity,

Water Resources Research, 54, 1688-1715, 2018.

Huang, G. H.: Model identifiability, Wiley StatsRef: Statistics Reference Online, 2005.

Merz, R., Parajka, J., and Bloschl, G.: Time stability of catchment model parameters: Implications for climate impact analyses, Water Resources Research, 47, 10.1029/2010wr009505, 2011.

Ouyang, Y., Xu, D., Leininger, T. D., and Zhang, N.: A system dynamic model to estimate hydrological processes and water use in a eucalypt plantation, Ecological Engineering, 86, 290-299, 10.1016/j.ecoleng.2015.11.008, 2016.

Pfannerstill, M., Guse, B., and Fohrer, N.: Smart low flow signature metrics for an improved overall performance evaluation of hydrological models, Journal of Hydrology, 510, 447-458, https://doi.org/10.1016/j.jhydrol.2013.12.044, 2014.

Wagener, T., and Kollat, J.: Numerical and visual evaluation of hydrological and environmental models using the Monte Carlo analysis toolbox, Environ Modell Softw, 22, 1021-1033, https://doi.org/10.1016/j.envsoft.2006.06.017, 2007.

Xiong, M., Liu, P., Cheng, L., Deng, C., Gui, Z., Zhang, X., and Liu, Y.: Identifying time-varying hydrological model parameters to improve simulation efficiency by the ensemble Kalman filter: A joint assimilation of streamflow and actual evapotranspiration, Journal of Hydrology, 568, 758-768, https://doi.org/10.1016/j.jhydrol.2018.11.038, 2019.

Zhang, D. J., Chen, X. W., Yao, H. X., and Lin, B. Q.: Improved calibration scheme of SWAT by separating wet and dry seasons, Ecol Model, 301, 54-61, 10.1016/j.ecolmodel.2015.01.018, 2015.

---

## Author Comment (AC2) · 10 Jan 2020

Dear Anonymous Referee #2,

Re: Manuscript # HESS-2019-544 entitled "Dynamics of hydrological model parameters: mechanisms, problems, and solution".

We are very grateful for the Referee's comments and encouragement. We have carefully studied and considered all comments in making revision and a point-by-point response is as follows. For clarity, all comments are given in black and responses are given in the blue text. The revised parts in our manuscript are highlighted in red.

Please also note the supplement to this comment:

https://www.hydrol-earth-syst-sci-discuss.net/hess-2019-544/hess-2019-544-AC2-supplement.pdf

[Figure]

**Supplement:**

**Replies to Referee #2**

Dear Anonymous Referee #2,

Re: Manuscript # HESS-2019-544 entitled "Dynamics of hydrological model parameters: mechanisms, problems, and solution".

We are very grateful for the Referee's comments and encouragement. We have carefully studied and considered all comments in making revision and a point-by-point response is as follows. For clarity, all comments are given in black and responses are given in the blue text. The revised parts in our manuscript are highlighted in red.

Yours sincerely,

Kairong Lin (Ph.D.)
Professor in hydrology
E-mail: linkr@mail.sysu.edu.cn

**Title: Dynamics of hydrological model parameters: mechanisms, problems, and solution**

Please address the following issues:
1 The introduction should have a paragraph stating the objectives of the paper.
**Reply:** A paragraph stating the objectives of the paper is *added* in the Introduction section of revision as suggested. The explicit information is as follows:

"This study aims to investigate the underlying causes of poor model performance in hydrological models with dynamic parameters via designing five calibration schemes, and explore the potential reasons for the poor response of the dynamic parameter set to the catchment dynamics are explored."

2 In the methodology, it is unclear what happens with the dynamic parameters during the validation period. Are they set to the same values as in the calibration period? Do they follow the exact same dynamics? Are the values dependent on the calendar day?
**Reply:** Thanks for the comment and sorry that we failed to state it clear enough in the previous version, and is now clarified. It is feasible that the dynamic parameters during the validation period are set to the same as in the calibration period in this study. The values are dependent on the calendar days. The reasons are as follows. Our previous research (Lan et al., 2018) focused on the reasonable sub-period clustering based on the dynamic catchment characteristics. The hydrological model was calibrated in each sub-period to achieve the dynamics of the parameter set. Namely, the calendar year is clustered into four sub-annual periods based on hydrological similarities. Most importantly, the clustering results are further verified by the hydrological data in the validation period. The study showed that the clustering results of the validation period are almost the same as the results of the calibration period. The reason is given that the selected study areas, which are the sub-basins of the Hanjiang River basin, are located in the monsoon region of the East Asia subtropical zone. The variations of both climate conditions and vegetation density and types are significantly seasonal (Fang et al., 2002). Hence, they are ideal places for studying the sub-period calibrations. The above discussion is supplemented in the *Methodology* section of the revision.

3 Correct the units in Table 1 (fluxes).
**Reply:** Revised as suggested in Table 1.

4 In the results, the term "model performance" is very generic. Can it be replaced with something more specific?
**Reply:** Thanks for the Reviewer's suggestion. The more specific explanation for "model performance" is supplemented in the *Results* section of revision. The detailed information is as follows:

"For a concise model evaluation, the model performance is analyzed with multi-metric frameworks with appropriate performance metrics, including five-segment evaluation (5FDC, flow duration curve with root mean square error) (Pfannerstill et al., 2014), Nash-Sutcliffe efficiency index (NSE) (Nash and Sutcliffe, 1970) and the logarithmic transformation (LNSE). For the robustness of model evaluation, the transferability of the optimized parameters between the calibration period and the validation period is considered."

**References:**

Fang, J. Y., Song, Y. C., Liu, H. Y., and Piao, S. L.: Vegetation-climate relationship and its application in the division of vegetation zone in China, Acta Bot Sin, 44, 1105-1122, 2002.

Lan, T., Lin, K. R., Liu, Z. Y., He, Y. H., Xu, C. Y., Zhang, H. B., and Chen, X. H.: A Clustering Preprocessing Framework for the Subannual Calibration of a Hydrological Model Considering Climate-Land Surface Variations, Water Resources Research, 54, 10,034-010,052, 10.1029/2018wr023160, 2018.

Nash, J. E., and Sutcliffe, J. V.: River flow forecasting through conceptual models part I — A discussion of principles, J Hydrol, 10, 282-290, 10.1016/0022-1694(70)90255-6, 1970.

Pfannerstill, M., Guse, B., and Fohrer, N.: Smart low flow signature metrics for an improved overall performance evaluation of hydrological models, J Hydrol, 510, 447-458, 10.1016/j.jhydrol.2013.12.044, 2014.